# Natural Red Pigment Production by *Monascus Purpureus* in Submerged Fermentation Systems Using a Food Industry Waste: Brewer’s Spent Grain

**DOI:** 10.3390/foods8050161

**Published:** 2019-05-11

**Authors:** Selim Silbir, Yekta Goksungur

**Affiliations:** 1Engineering Faculty, Department of Food Engineering, Iğdır University, Iğdır 76000, Turkey; selim.silbir@gmail.com; 2Engineering Faculty, Department of Food Engineering, Ege University, Izmir 35040, Turkey

**Keywords:** Natural red pigment, *Monascus purpureus*, Brewer’s spent grain, Submerged fermentation, Plackett-Burman design, Chemical characterization, X-ray photoelectron spectroscopy (XPS), Fourier-transform infrared spectroscopy (FTIR)

## Abstract

This paper studies the production of natural red pigments by *Monascus purpureus* CMU001 in the submerged fermentation system using a brewery waste hydrolysate, brewer’s spent grain (BSG). The chemical, structural and elemental characterization of the BSG was performed with Van-Soest method, Fourier-transform infrared spectroscopy (FTIR) and X-ray photoelectron spectroscopy (XPS), respectively. The lignocellulosic structure of BSG was hydrolyzed with a dilute sulfuric acid solution (2% (*w*/*v*)) followed by detoxification with Ca(OH)_2_. Maximum red pigment production (22.25 UA_500_) was achieved with the following conditions: 350 rpm shake speed, 50 mL fermentation volume, initial pH of 6.5, inoculation ratio of 2% (*v*/*v*), and monosodium glutamate (MSG) as the most effective nitrogen source. Plackett–Burman design was used to assess the significance of the fermentation medium components, and MSG and ZnSO_4_·7H_2_O were found to be the significant medium variables. This study is the first study showing the compatibility of BSG hydrolysate to red pigment production by *Monascus purpureus* in a submerged fermentation system.

## 1. Introduction

In recent years, the food industry has focused on the production of natural pigments from plants and microbial sources to overcome the use of synthetic pigments which are potentially hazardous to human health and the environment. Natural pigments produced by microorganisms have gained more importance because of their low water solubility and the unstable nature of plant-derived pigments against heat and light [1]. Monascus pigments have been used as a natural coloring agent and natural food additive in East Asia. These pigments, which are produced by various species of *Monascus*, improve the color of foods and their sensory characteristics [2,3]. Monascus pigments also have applications as pharmaceuticals since they are reported to have health benefits. Rice fermented with *Monascus purpureus* was found to be effective for the management of cholesterol, diabetes, cardiovascular disease, and also for the prevention of cancer [4].

*Monascus* is a xerophilic fungus which grows in a wide variety of natural substrates including rice and other cereals [5]. The genus *Monascus* is divided into three species: *purpureus, ruber*, and *pilosus*, which are mainly isolated from oriental foods. The most important characteristic of this genus is their ability to synthesize pigments from polyketide chromophores and β-keto acids by esterification. The pigments produced by *Monascus purpureus* are classified into at least six types of pigments based on color: (1) red pigment (rubropunctamin, C_21_H_26_NO_4_, and monascurubramin, C_23_H_27_NO_4_); (2) orange pigment (rubropunctatin, C_21_H_22_O_5_ and monascorubrin, C_23_H_26_O_5_) and (3) yellow pigment (monascin, C_21_H_26_O_5_ and ankaflavin, C_23_H_30_O_5_). The structure of pigments produced by *Monascus* species depends on factors such as the type of substrate and nitrogen source, pH, temperature, and agitation [2,3,6].

There are various studies on the production of Monascus pigments from food industrial wastes like potato powder [1], bakery waste [2], jackfruit seed [7], grape waste [8], sugarcane bagasse [9,10], wheat [11], sweet potato [12], and prickly pear juice [13].

Brewer’s spent grain (BSG) is the most important waste generated by the breweries and corresponds to 85% of the total waste generated in beer production [14]. Since BSG has a low price and is abundant, these properties contribute to the economy of any process that uses this waste biomass. Hence, it will be beneficial to utilize BSG as a substrate in pigment production. There are not many studies on the use of BSG in pigment production in the literature. There is only one paper about pigment production using BSG and it is about solid-state fermentation. Babitha et al. [7], investigated the feasibility of some agro-industrial residues for the production of pigments by *Monascus purpureus*, including brewer’s spent grain as a substrate in solid state fermentation. They obtained the best results with jackfruit seed powder and selected this substrate for subsequent studies.

Thus, this is the first study that investigates in detail the use of nutrient-rich brewer’s spent grain-derived hydrolysate in the production of red pigments from *Monascus purpureus* CMU-001 in submerged fermentation. 

## 2. Materials and Methods 

### 2.1. Microorganism and Media

*Monascus purpureus* CMU001 was supplied by Professor Saisamorn Lumyong from Chiang Mai University, Department of Biology. The culture was maintained and sporulated on potato dextrose agar (Merck, Germany). The semi-synthetic fermentation medium was a modification of Silveira et al., [8] and consisted of (g/L): MSG, 8; KH_2_PO_4_, 5; K_2_HPO_4_, 5; MgSO_4_·7H_2_O, 0.01; CaCl_2_, 0.01; ZnSO_4_·7H_2_O, 0.01; and liquid brewer’s spent grain hydrolysate as a carbon source.

Brewer’s spent grain (BSG) was supplied by Turk Tuborg Bira ve Malt Sanayi A.Ş. BSG was washed to remove residual starch and its pH neutralized and dried in an oven at 65 °C to 4% moisture content. The substrate was then milled with a pilot scale hammer mill to increase the surface area for better acidic hydrolysis and stored in airtight jars until use. 

### 2.2. Preparation of BSG Hydrolysate

Previously washed, dried and milled brewer’s spent grain was treated with different concentrations of sulfuric acid (1–6% (*w*/*v*)) in different solid (BSG): liquid(dilute sulfuric acid) ratios (1:6–10 (w:w)) at 120 °C for 15 min. After hydrolysis, the supernatant was recovered by centrifugation at 6000 rpm for 10 min. 

In order to reduce the inhibitory substances generated during acid hydrolysis, a detoxification procedure (overliming) was applied to the raw hydrolysate as recommended by Carvalheiro et al. [15]. The pH of the raw hydrolysate was increased to 10 with Ca(OH)_2_ and held at 55 °C for 1 h. After centrifugation, the pH of the supernatant was adjusted to pH 5.5 with 25% (*w*/*w*) H_2_SO_4_ and used as brewer’s spent grain-based fermentation medium.

### 2.3. Cultivation and Fermentation Conditions

The strain was maintained on potato dextrose agar (PDA) dishes, stored at 4 °C, and transferred every 4 weeks to fresh PDA slants incubated at 30 °C for 7 days. Spore solution containing 1.0 × 10^6^ spores/mL was collected using sterile distilled water. Fifty milliliters of semi-synthetic BSG fermentation media was inoculated with 2 % (*v*/*v*) of spore solution. Fermentation experiments were done in a rotary shaker incubator (Sartorius Stedim, Certomat BS-1, Germany) operated at 350 rpm, 30 °C for 7 days. Biomass growth and pigment production were determined at equal time intervals. Different initial fermentation volume (25–75 mL), initial pH (5.5–7.5), inoculation ratio (1–4% (*v*/*v*)), and nitrogen sources (monosodium glutamate, malt sprouts, corn steep liquor, peptone, urea and yeast extract) at a nitrogen concentration equivalent to 8 g/L MSG were tested for Monascus pigment production. 

### 2.4. Analytical Methods

Brewer’s spent grain samples were analyzed on the contents of cellulose, hemicellulose, lignin, minerals, and total soluble mass using the Van Soest method [16]. Total solids/Volatile solids (TS/VS) analysis was done to determine the content of total solids, volatile solids and ash according to the standard method 2540 [17]. 

The infrared spectra of milled brewer’s spent grain were recorded between 4000 and 698 cm^−1^ at a resolution of 4 cm^−1^ using Carry 660 FTIR spectrometer (Agilent Technologies, Santa Clara, CA, USA) mounted with a single reflection diamond MIRacle attenuated total reflectance (ATR) accessory a high-pressure clamp (Pike Technologies, Fitchburg, WI, USA). One-hundred-and-twenty-eight scans were co-added for each spectrum to improve the signal-to-noise ratio. The powder was directly placed on a single reflection diamond ATR and pressed with a pressure clamp for having a good contact between the crystal and sample. Four spectra of samples were collected by using a Varian Resolutions Pro 4.05.

X-ray photoelectron spectroscopy (XPS) (Thermo Scientific, K-Alpha XPS, Waltham, MA, USA) was used to determine the elemental composition (C, N, O, P%) of the milled raw material by sending an analytical spot diameter of 200 µm monochromatic Al K-α X-rays (1486.68 eV) to the surface of the sample. Survey and high-resolution spectra of Fe 2p, P 2p, O 1s, N 1s, and C 1s core levels were recorded with a constant pass energy of 200 eV and 50 eV, respectively. A take-off angle of 90° was used in the experiments. In order to detect surface components, XPS measurements were made at different points of the sample after the survey spectra. 

The Kjeldahl method [18] was used to determine crude protein content as a function of nitrogen content.

To determine the amount of red pigment extracellularly produced by *M.purpureus*, the mycelia were separated from the fermentation broth using Whatman # 3 filter paper. The filtrate was then centrifuged using a centrifuge (Hettich Universal 320 R, Andreas Hettich GmbH & Co. KG, Tutlingen, Germany). The supernatant was collected to calculate the extracellular pigment concentration by a UV-Vis spectrophotometer (Thermo, Genesys 10S UV-Vis, Waltham, MA, USA) at 500 nm wavelength. Dilution factors were taken into consideration and values obtained were expressed as specific absorbance units (UA_500_) [8].

Cell growth was evaluated gravimetrically. *Monascus* mycelia were separated and dried to a constant weight at 65 °C and weighed on an analytical scale. 

The Dinitrosalicylic(DNS) colorimetric method [19] was used to find the reducing sugar content of the fermentation medium. The reported data are the average values ± standard deviations of three replicates.

### 2.5. Statistical Analysis

Plackett Burman Design (PB) [20] was applied for screening and understanding the role of various nutrient components in red pigment production using Design Expert Statistical Software (Release 11.1.0.1). MSG, KH_2_PO_4_, K_2_HPO_4_, MgSO_4_·7H_2_O, CaCl_2_, and ZnSO_4_·7H_2_O were tested (Table 1) for their effect on red pigment production and the ones with statistically negative or without effect on the red pigment production were excluded from the fermentation medium. The significance of the six examined parameters was evaluated by *p*-value (Table 2). Parameters with a *p*-value smaller than 0.05 were considered statistically significant. 

Plackett-Burman experimental design and the range of variable levels are given in Table 1. There were 12 trials in the experimental design in which low (−) and high (+) levels of each medium component were studied. The following polynomial model explains the relationship between produced red pigment (UA_500_); Y and fermentation medium components (Equation (1));
(1)$$Y=\;\alpha_{0}+\;\overset{n}{\underset{i=1}{\sum }}\alpha_{i}X_{i}$$


In Equation (1), *Y* is the dependent variable (response), α_i_ is the regression coefficient for linear effects, α_0_ is the regression coefficient for the intercept, and *X_i_* is the coded independent variable. 

## 3. Results and Discussion

### 3.1. The Chemical Composition and Characterization of Brewer’s Spent Grain(BSG)

#### 3.1.1. Chemical Composition

The BSG used for pigment production was a lignocellulosic substrate consisting of hemicellulose (53.1%), cellulose (19.2%) and lignin (8.5%), with smaller proportions of ash (3.68%) and nitrogen (2.76%) as shown in Table 3. 

It was stated in the literature that the composition of brewer’s spent grain is variable. Xiros et al. [21] found that BSG contained 19–40% and 9–25% of hemicellulose and cellulose, respectively. Mussatto and Roberto [22] also found that hemicellulose and cellulose were the main components in BSG followed by lignin. Carvalheiro et al. [23] found that glucan and xylan were the main polysaccharides present in BSG. BSG is a massive waste of barley obtained after the mashing step in the brewing industry. The major constituent of BSG is barley husk which is lignocellulosic in structure. The period of harvest, variety, the mashing and malting conditions employed, and the kinds of supplements added during the brewing process affect the composition of the barley [22]. 

#### 3.1.2. Fourier-Transform Infrared Spectroscopy (FTIR)

The infrared spectrum (FTIR) of the milled brewer’s spent grain is shown in Figure 1. The broad stretching intense peak at around 3313 cm^−1^ represents hydroxyl and amine groups as shown in the spectrum. The band at 2927 cm^−1^ is related to the asymmetric stretch (nC–H) of–CH2 groups and the corresponding symmetric stretch can be found at 2858 cm^−1^. The peak at 1743 cm^−1^ can be related to ester bonds or carboxylic (C=O stretching vibration) linkages in lignin and hemicellulose, which is generally evident in untreated BSG and diminishes according to the effectiveness of the pretreatment [24]. Protein-based bonds of nC=O amide I (1635 cm^−1^) [25] were observed. The peak at 1523 cm^−1^ corresponds to C=C bonds in the aromatic ring of lignin. The peak at 1458 cm^−1^ represents the C–H asymmetric deformation vibrations of aromatic skeletal in lignin [26]. The peak at 1361 cm^−1^ may be associated with highly conjugated C=O stretching vibrations in carboxylic groups. The bands at 1238 cm^−1^ and 1033 cm^−1^ correspond to C–H stretching vibrations, which is characteristic of cellulose content in BSG [24,26]. Parallel results were also found by other researchers [24,26] who used infrared spectroscopy to characterize the chemical structure of BSG by identifying the functional groups present in the samples. 

#### 3.1.3. X-ray Photoelectron Spectroscopy (XPS)

The elemental composition of milled brewer’s spent grain samples was examined with XPS. According to the results of XPS analysis, BSG samples contained elements of carbon, oxygen, nitrogen, and phosphorus (Table 4 and Figure 2). The signals from carbon, oxygen, nitrogen, and phosphorus elements are summarized in Table 4. C–C* single bonds (284.99 and 285.3 eV) and C*–OH and C–O–C* single bonds (286.09 eV) indicate the presence of cellulose and hemicellulose in the milled BSG samples. These structures are also supported by signals from the oxygen element of 532.86 eV, 532.4 eV and 532.92 eV which corresponds to the C=O * double bond, C–O*–H single bond and C–O*–H single bond, respectively. The peak of the C=C double bond at 284.36 eV indicates the presence of lignin in brewer’s spent grain. The nitrogen signals (C-N*) at 400.11 and 400.44 eV originate from the amine groups in the protein structure of the samples. According to these results and literature data, BSG contains cellulose, hemicellulose, lignin, and protein [26,27].

### 3.2. Effect of Acid Concentration

The BSG used in this work was a lignocellulosic material composed of hemicellulose (53.1%), cellulose (19.2%) and lignin (8.5%) as shown in Table 3. Hence, a pretreatment step is necessary for the disruption of the lignocellulosic structure of BSG and the release of fermentable monosaccharides like glucose, arabinose, and xylose. Dilute acid hydrolysis is a rapid and basic method employed in the hydrolysis and pretreatment of lignocellulosic substrates such as BSG [15]. The effect of acid concentration on the production of Monascus red pigment was determined using different sulfuric acid concentrations (1–5% (*w*/*v*)). The solid: liquid ratio of 1:8 (w:w), 120 °C and 15 min were used for the pretreatment step. The hydrolysate was then used for pigment production with *Monascus purpureus* CMU001. The fermentation experiments were done in shake flask cultures at 30 °C, pH 5.5 for 7 days. The highest pigment production of 15.35 UA_500_ was obtained in the fermentation medium containing BSG hydrolyzed with 2% (*w*/*v*) sulfuric acid as shown in Figure 3A. When the acid concentration was increased, a gradual decrease in pigment formation was observed. The decrease in pigment formation observed under extreme hydrolysis conditions was probably a result of the production of inhibitory compounds such as HMF and furfural which are known to have toxic effects on microorganisms. In addition, low pigment production (3.07 UA_500_) was observed in BSG medium hydrolyzed with 1% (*w*/*v*) sulfuric acid. This was probably due to the ineffective disruption of the lignocellulosic structure and inadequate release of fermentable sugars. 

This showed that *Monascus purpureus* did not synthesize cellulase to utilize the cellulose moiety of BSG. The rigid lignocellulosic structure of BSG also prevented the utilization of cellulose and therefore pigment production by *Monascus purpureus*.

### 3.3. Effect of Solid: Liquid Ratio

Three different solid: liquid ratios ((1:6), (1:8) and (1:10) (w:w)) of brewer’s spent grain: dilute sulfuric acid were used to hydrolyze BSG using 2% (*w*/*v*) dilute sulfuric acid. BSG hydrolysates obtained were then used for fermentation experiments with *M. purpureus* in shake flask cultures. BSG hydrolyzed with 2% (*w*/*v*) H_2_SO_4_ using (1:6) (w:w) solid: liquid ratio at 120 °C for 15 min gave the highest value of red pigment production (16.75 UA_500_) as shown in Figure 3B. Slightly lower pigment production values of 15.35 and 13.10 UA_500_ were obtained with media hydrolyzed at (1:8) and (1:10) solid: liquid ratios, respectively. Solid: liquid ratios lower than (1:6) could not be used due to operational difficulties. These results proved that all the hydrolysates prepared by dilute acid hydrolysis in (1:6), (1:8) and (1:10) (w:w) solid: liquid ratios showed good fermentation characteristics when utilized as a fermentation medium for the red pigment formation by *Monascus purpureus.*

Musatto and Roberto [28] obtained the highest xylitol production when BSG was hydrolyzed with 100 mg H_2_SO_4_/g dry substrate. The solid: liquid ratio and the reaction time of sulfuric acid pretreatment was [1:8] (w:w) and 17 min, respectively. Mussatto et al. [14] also used sulfuric acid for BSG hydrolysis prior to lactic acid production by *Lactobacillus delbrueckii*. They employed a sulfuric acid (H_2_SO_4_) solution of 1.25% in a solid: liquid (g:g) ratio at 120 °C/17 min. 

### 3.4. Effect of Shaking Speed and Medium Volume on Pigment Synthesis

Shaking is crucial in overcoming mass transfer resistances in aerobic fermentation systems and the oxygen transfer rate is directly related with shaker speed [29]. In the present study, the effect of shaking speed and liquid volume on pigment production by *M. purpureus* was studied in BSG medium. Fermentation trials were done in a rotary shaker incubator operated at 250, 300, 350, and 400 rpm. BSG was pretreated with 2% (*w*/*v*) sulfuric acid solution with a solid: liquid ratio of 1:6 (*w*/*w*) to prepare the fermentation medium in all experiments. Overliming (Ca(OH)_2_) was used as the detoxification method. Fermentation lasted for 7 days in 250-mL Erlenmeyer flasks with 50 mL of fermentation medium at pH 5.5. Pigment production was increased with the increased shaking speed (from 250 to 350 rpm) and decreased afterward (Figure 3C). At the lowest shaking speed value of 250 rpm, larger pellets of *Monascus purpureus* were observed, indicating mass diffusion problems. The highest red pigment production of 16.75 UA_500_ was obtained at a shaker speed of 350 rpm. The decrease in red pigment production at a shaker speed of 400 rpm (13.68 UA_500_) might be due to the shear-sensitive characteristics of *Monascus* mycelia. 

To determine the influence of the medium volume in the Erlenmeyer flask on the production of red pigments: 25, 50 and 75 mL of fermentation medium were used in a rotary shaker incubator operated at 350 rpm shaking speed. The highest pigment formation of 16.75 UA_500_ was observed in 50 mL fermentation volume (Figure 3D). The decrease of pigment production in 25 and 75 mL fermentation media volumes might be due to the lower gas–liquid mass transfer area resulting in lower oxygen transfer to fermentation liquid and finally to *Monascus* mycelia. Low pigment production of 8.18 UA_500_ in 25 mL medium volume might result from the inefficient shaking of the fermentation medium. The fermentation medium did not rotate well enough with the movement of the orbital shaker which led to lower volumetric mass transfer rates [30]. Vortex formation was noticed in 75 mL of fermentation medium that resulted in the poor mass transfer of oxygen and/or substrate, causing a decrease in red pigment production. Silveira et al. [8] also reported that low oxygen partial pressure decreased *Monascus* pigment production in submerged culture experiments.

### 3.5. Effect of Initial pH

pH is a significant factor in the activation of important enzymes in pigment production by *Monascus purpureus* [31]. Various initial pH values (5.5, 6.0, 6.5, 7.0 and 7.5) were tested to observe the effect of pH on *M. purpureus* red pigment synthesis. BSG hydrolyzed with 2% (*w*/*v*) sulfuric acid with a solid: liquid ratio of 1:6 (*w*/*w*) was used to prepare BSG-based fermentation medium. Fermentation experiments were performed at 30 °C for 7 days. The pH used after sterilization was used as the initial pH since the pH of the medium changed after sterilization. The change in pH after autoclaving might result from the buffering effect of the substrate (brewer’s spent grain) or salt solutions.

Initial pH 6.5 gave the highest pigment concentration of 22.25 UA_500_. Slightly lower red pigment values of 21.38 and 20.43 UA_500_ were obtained at an initial pH of 6.0 and 7.0, respectively (Figure 3E). Lower red pigment production values of 16.75 and 12.98 UA_500_ were obtained at pH values of 5.5 and 7.5, respectively. Our results showed that the synthesis of red pigment by *M. purpureus* slows down with increasing or decreasing pH values at our experimental conditions. 

The pH of the medium strongly affects red pigment production in *M. purpureus* since red pigments (extracellular and water soluble) are produced by the chemical modification of orange pigments under relatively higher pH values in the presence of a suitable nitrogen source [32]. Parallel to the findings of our study, many researchers showed that the pH of the fermentation medium had an important effect on pigment synthesis by *M. purpureus*. [31,32,33].

### 3.6. Effect of Inoculation Ratio

The biomass and pigment concentration of *Monascus* mycelia are affected by the initial inoculum concentration. To observe the effect of inoculum concentration on pigment production, 50 mL of BSG fermentation medium was inoculated with 1, 2, 3, and 4% (*v*/*v*) spore suspension solution, corresponding to 0.5 × 10^6^, 1.0 × 10^6^, 1.5 × 10^6^, and 2.0 × 10^6^ spores/50 mL of fermentation medium. The highest pigment production (22.25 UA_500_) was observed in the pretreated BSG medium inoculated with 2% (*v*/*v*) spore suspension. The pigments produced in the fermentation media inoculated with 1, 3 and 4% spore suspensions were 15.87, 18.15 and 11.36 UA_500_, respectively (Figure 3F). Our results showed that the low inoculum ratio reduced the amount of biomass leading to a lower concentration of pigment. However, the high inoculum ratio yielded a high biomass concentration that resulted in rapid consumption of nutrients in the fermentation medium required for pigment synthesis. 

It is well known that spore inoculum concentration affects growth, morphology, volumetric productivity, and enzymes of fungi propagated in submerged culture [34]. However, there are few reports about the effect of inoculum ratio or inoculum size on *Monascus* growth and product formation characteristics. Babitha et al. [7] produced *Monascus* pigments from jackfruit seed by solid-state fermentation and observed poor pigment production at lower and higher inoculum levels similar to the results of this study. They obtained the highest pigment production with an inoculum size of 3 mL (9 × 10^4^ spores/gram dry substrate). 

### 3.7. Effect of Nitrogen Source

The nitrogen source is essential in red pigment production by *Monascus* species since reactions with amino group-containing compounds promote water-soluble, extracellular red pigment production. Monosodium glutamate is a favorable nitrogen source for *M. purpureus* and has been documented by many authors [8,31,32,35,36].

Although MSG is essential in red pigment production, its high cost limits its industrial use. To decrease the cost of pigment production, alternative nitrogen sources were replaced by MSG (8 g/L) on an equivalent nitrogen basis. Fermentation experiments were done in shake flasks inoculated with 2% (*v*/*v*) spore suspension solution at 30 °C, pH 6.5 for 7 days. The batch of each source used was equivalent to a nitrogen amount of 8.0 g/L MSG. The different sources of nitrogen and concentrations used were (g/L): yeast extract (YE), 5.81. peptone (PEP), 4.07; corn steep liquor (CSL), 8.83; malt sprouts (MS), 10.69; and urea (URE), 1.30. Results demonstrated that the nitrogen source greatly influenced the red pigment formation by *M. purpureus*. The highest pigment formation (22.25 UA_500_) was achieved when MSG was used (Figure 3G). *M. purpureus* produced 16.46 and 13.02 UA_500_ of red pigment when CSL and yeast extract were used as the nitrogen source, respectively. *M. purpureus* also utilized other nitrogen sources; however, lower red pigment production values were obtained. Our results showed that corn steep liquor was a promising alternative source for the production of natural pigments by *M. purpureus*. However, further research is needed to improve red pigment production when CSL is used as the nitrogen source.

### 3.8. Selecting the Important Medium Components by Plackett-Burman Design

In this research, Plackett–Burman design was employed to test different nutritional variables in pigment production. The aim was to achieve maximum red pigment formation by *M. purpureus*. The six variables tested were MSG, K_2_HPO_4_, KH_2_PO_4_, MgSO_4_·7H_2_O, CaCl_2_, and ZnSO_4_·7H_2_O under submerged fermentation. All six factors chosen in the study were tested at two levels, namely low level (−1) and high level (+1), and the experimental runs are given in Table 1. Table 2 shows the estimated regression coefficients, main effect, and *p* and *t* values. Among the chosen variables, MSG and ZnSO_4_·7H_2_O were the significant medium components (*p* < 0.05) (Table 2). Regression coefficients in coded units were used to establish a simple polynomial model in order to estimate pigment production (Equation (2));
Red pigment (UA_500_) = +13.86 − 3.48*A − 1.69*B + 1.42*C−0.017*D − 0.21*E + 2.94*F(2)


The results obtained in the PB experiment are in agreement with the literature. Bau and Wong [37] investigated the effect of zinc on growth, pigment formation and antibacterial activity of *M. purpureus*. They found that growth, antibacterial activity and pigment production of *M. purpureus* were highly affected by adding zinc to the fermentation medium. They stated that zinc inhibits the growth of *M. purpureus* and increases the production of pigments and antimicrobials. They concluded that zinc might have three roles in *Monascus* metabolism which were promoting glucose uptake, inhibiting mycelial growth and stimulating pigment production. 

PB design was also used by other researchers in Monascus pigment production. Sharmila et al. [1] used PB design to evaluate the significant medium variables in Monascus fermentation. They stated that K_2_HPO_4_, ZnSO_4_·7H_2_O and MSG together with potato powder were significant variables for pigment synthesis by *M. purpureus*. Prajapati et al. [31] used PB design to identify the medium components which affect the pigment formation by *M. purpureus*. They found that tryptone, glucose and pH were significant factors among various variables screened. 

### 3.9. Kinetics of Red Pigment Synthesis by Monascus Purpureus

The kinetics of *M. purpureus* growth together with pigment synthesis were examined in the BSG medium. The fermentation medium was prepared by the dilute sulfuric acid hydrolysis of BSG followed by a detoxification (overliming-unpublished data) step. The medium consisted of dilute sulfuric acid hydrolyzed BSG, MSG (8 g/L) and ZnSO_4_·7H_2_O (0.01 g/L), which were significant medium variables selected from PB screening experiments. Fermentation trials were done in shake flasks at 30 °C, pH 6.5 with a spore inoculation ratio of 2% (*v*/*v*) which corresponds to 2 × 10^4^ spores/mL fermentation medium. The highest pigment production of 22.25 UA_500_ was obtained on the 7th day of fermentation, and afterward it decreased (Figure 4). The oxidative enzymes of the microorganism might be responsible for the oxidation of the pigment, which resulted in a concentration decrease especially during the late period of fermentation. Sharmila et al. [1] obtained the highest pigment production of 6.94 ODU/mL at the 6th day of fermentation by using potato powder as the carbon source. Meinicke et al. [36] obtained 7.38 UA_480 nm_ of pigment formation after 7 days of fermentation using glycerol as the carbon source. Hamano and Kilikian [35] produced red pigments from a complex culture medium composed of glucose and obtained 20.7 U of pigment production. Haque et al. [2] produced 24 AU/g glucose of pigment using bakery waste hydrolysate. Orozco and Kilikian [33] produced 11.3 U of red pigment from glucose as the carbon source. The variation in the literature data for Monascus pigment production may be related to several factors such as the strain of microorganism, the type of substrate and nitrogen source, the fermentation system, the method of pigment estimation, and the conditions used during fermentation. 

On the first day of fermentation, there were no biomass and no pigment formation. First, *Monascus purpureus* mycelia and red color were observed on the second and third day of the fermentation respectively. Maximum pigment production was obtained on day 7 (Figure 5). A maximum biomass concentration of 5.73 g/L was observed after 7 days of fermentation and then it declined. The reducing sugar concentration decreased during fermentation and almost depleted on the 5th day of fermentation. The microorganism probably used non-reducing hydrolysis products of BSG as the carbon source after the 5th day of fermentation. 

The pH of the fermentation broth decreased slightly during the first 5 days of fermentation from an initial value of 6.5 to 5.89 and then increased slowly up to 7.74 at the end of fermentation. This was due to the deamination of amino acids present in the medium by *M. purpureus* and the production of ammonia, which increased the pH of the fermentation broth. 

## 4. Conclusions

Monascus pigments have received worldwide attention for their multiple health benefits; they appear to have anti-mutagenic, anti-cancer, anti-obesity, anti-inflammation, anti-diabetes, and cholesterol-lowering mechanisms. This research article is the first on the evaluation of hydrolyzed and detoxified brewer’s spent grain for red pigment production by *Monascus purpureus* in submerged fermentation systems. This paper suggests an innovative approach of using waste produced in large amounts by breweries and contributes to the production of natural pigment products which have multiple benefits for health. Further studies will be focused on the scale-up and stability modeling of the pigments produced. 

## Figures and Tables

**Figure 1 foods-08-00161-f001:**
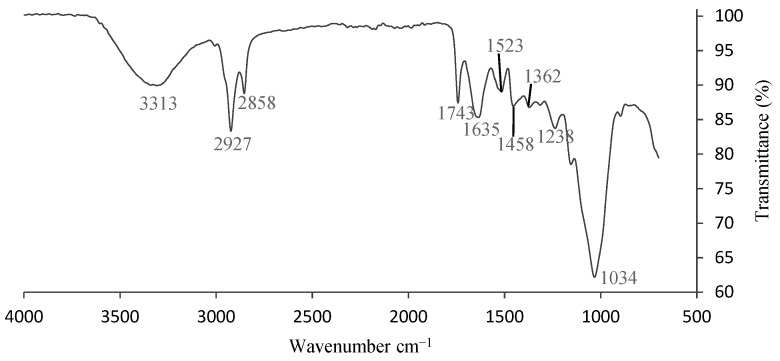
Infrared spectrum(FTIR) of milled brewer’s spent grain (BSG).

**Figure 2 foods-08-00161-f002:**
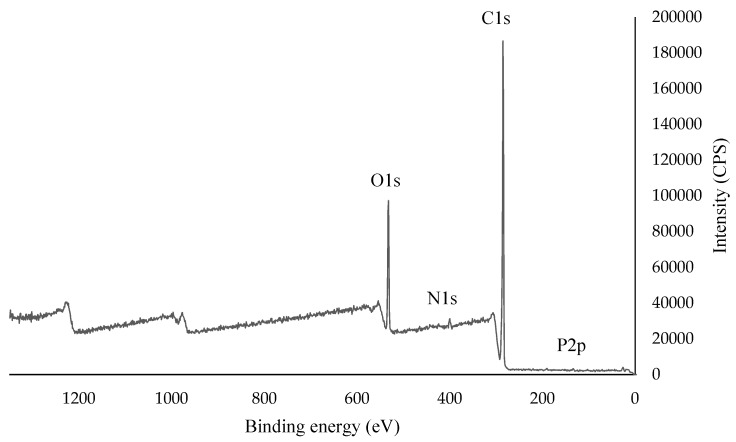
X-ray photoelectron spectroscopy (XPS) survey spectra of milled brewer’s spent grain.

**Figure 3 foods-08-00161-f003:**
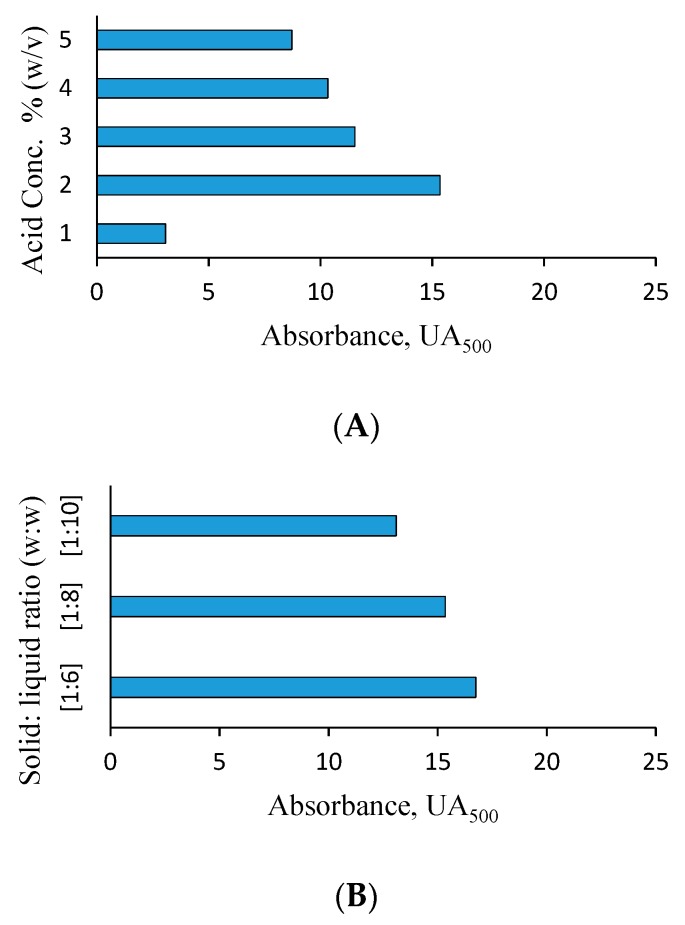
Effect of various parameters on red pigment production by *Monascus purpureus* CMU 001. (**A**, acid concentration; **B**, solid: liquid ratio; **C**, shaking speed; **D**, medium volume; **E**, initial pH; **F**, inoculation ratio; **G**, nitrogen source (based on equivalent nitrogen amount), (YE, yeast extract; URE, urea; PEP, peptone; CSL, corn steep liquor; MS, malt sprouts; and MSG, monosodium glutamate)). The standard deviation (SD) of each experimental point ranged from ±1.2 to ±4.8.

**Figure 4 foods-08-00161-f004:**
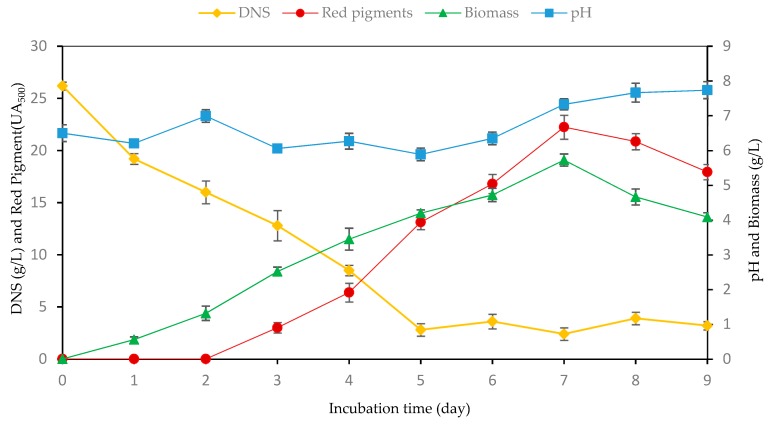
Batch fermentation profile showing red pigment “circles” (UA_500_,), biomass “triangles”, pH “squares”, and residual sugar “diamonds” for pigment production by *Monascus purpureus* CMU 001. (DNS: dinitrosalicylic colorimetric method; Fermentation conditions: 350 rpm, 30 °C, pH 6.5).

**Figure 5 foods-08-00161-f005:**
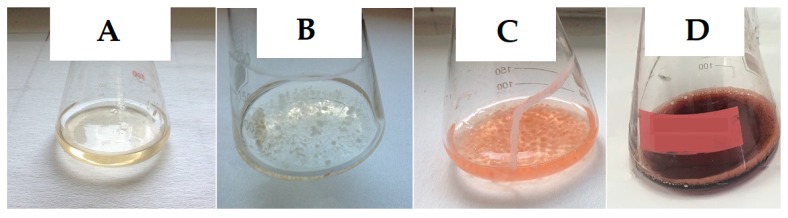
The profile of Monascus mycelia and the produced red pigment in the Erlenmeyer flasks during fermentation (**A**, 1st day; **B**, 2nd day; **C**, 3rd day; **D**, 7th day).

**Table 1 foods-08-00161-t001:** Level of variables and experimental design matrix for PB experiment with an observed response.

Code	Variable	Low Level (−)	High Level (+)	#	A	B	C	D	E	F	Pigment Production (UA_500_)
A	MSG	800	8000	1	−	−	+	−	+	+	21.76
B	K_2_HPO_4_	500	5000	2	+	−	+	+	−	+	17.39
C	KH_2_PO_4_	500	5000	3	−	+	+	−	+	+	21.25
D	MgSO_4_·7H_2_O	10	1	4	+	+	−	−	−	+	7.14
E	CaCl_2_	10	1	5	+	−	−	−	+	−	12.94
F	ZnSO_4_·7H_2_O	10	1	6	−	+	−	+	+	−	5.71
				7	+	+	−	+	+	+	13.94
				8	+	−	+	+	+	−	6.30
				9	−	−	−	−	−	−	15.62
				10	−	−	−	+	−	+	19.32
				11	+	+	+	−	−	−	4.58
				12	−	+	+	+	−	−	20.41

PB: Plackett Burman Design; UA_500_: absorbance units at 500 nm wavelength.

**Table 2 foods-08-00161-t002:** Estimated effects and coefficients (in actual units) for pigment production (UA_500_).

Term	Effect	Coeff	*t*	*p*	% Contribution
Constant		15.146	5.93	0.0000	
A) MSG	−6.96	−9.67	14.38	0.0192	31.57
B) K_2_HPO_4_	−3.38	−7.51	3.39	0.1395	7.44
C) KH_2_PO_4_	2.84	6.3 × 10^−4^	2.38	0.1976	5.23
D) MgSO_4_·7H_2_O	−3.5 × 10^−2^	−3.89 × 10^−3^	3.63 × 10^−4^	0.9857	7.97 × 10^−4^
E) CaCl_2_	−0.43	−0.04744	0.054	0.8276	0.12
F) ZnSO_4_·7H_2_O	5.87	0.6526	10.22	0.0330	22.45

*R*^2^ = 0.9122.

**Table 3 foods-08-00161-t003:** Composition of the milled brewer’s spent grain.

Compound	Amount (%)	Analysis/Method
HEMI	53.09	Van Soest [16]
CELL	19.24	
LIGN	8.53	
SOLU	19.15	
Total	100	
Ash	3.68	TS/VS [17]
TS	94.53	
VS	90.05	
Nitrogen	2.76	Kjeldahl [18]
Protein	17.25 *	
Carbon	81.63 ± 1.53	XPS
Nitrogen	2.00 ± 0.59	
Oxygen	15.99 ± 1.23	
Phosphorus	0.66 ± 0.08	

***** Protein content determined by multiplying N% with 6.25 conversion factor. HEMI, Hemicellulose; CELL, Cellulose; LIGN, Lignin; SOLU, Soluble solids; TS, Total solids; VS, Volatile solids; XPS, X-ray photoelectron spectroscopy.

**Table 4 foods-08-00161-t004:** XPS binding energy signals of carbon, oxygen, nitrogen and phosphorus elements.

Element	Binding Energy (eV)	Functional Group
P2p	134.1	P*-O, Phosphate
C1s	285.3	C-C*
N1s	400.11	C-NH_2_
O1s	532.92	C-O*-H
C1s	284.99	C-C*
C1s	284.36	C=C*
C1s	286.09	C-OH, C-O-C, C-N
C1s	288.26	C=O
N1s	400.44	C-NH_2_
O1s	532.4	C-O*-H
O1s	532.86	C=O*
P2p	133.25	P*-O, Phosphate

XPS: X-ray Photoelectron Spectroscopy. *: excited state.

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
