# Peer review of "Natural Red Pigment Production by Monascus Purpureus in Submerged Fermentation Systems Using a Food Industry Waste: Brewer’s Spent Grain"

_foods, 2019, doi:10.3390/foods8050161_

Round 1
Reviewer 1 Report
The reviewed paper at title:
Natural Red Pigment Production by Monascus purpureus in Submerged Fermentation Systems Using a Food Industry Waste: Brewer’s Spent Grain
This article suggests an innovative approach to the use of waste produced in large quantities by breweries and contributes to the production of natural pigment products that have many health benefits. Monascus pigments have received global attention due to the numerous health benefits that seem to have antimutagenic, anti-cancer, anti-obesity, anti-inflammatory, anti-diabetic and cholesterol-lowering mechanisms.
In my opinion, it requires some modifications. Additionally, several questions should be answered by the authors in detail, as many important issues are described too superficially:
Line 51-56: It's good to characterize Monascus purpureus ! This is a species of mold that has a purple-red color. During growth, this species breaks down the starch substrate into several metabolites, including pigments produced as secondary metabolites. The structure of pigments depends on the type of substrate and other specific factors during cultivation, such as pH, temperature and humidity. It produces many statins. The related fungi M. ruber and M. pilosus are also used in industrial applications.
Line 222-224: A very interesting observation, I think it needs a better discussion.
Line 233-235: I understand, therefore, that the best growing medium is a liquid substrate.
Line 251-255: Pretty confusing text regarding the speed of shaking.
Line 260-267: It is clear from the text that, in general, the method is quite unpredictable regarding the efficiency of pigment production.
Line 286-302: The text needs to be clarified, now it is quite unclear.
Line 362-365: These differences, however, are very large.
Line 381-384: Probably ? Certainly, these are the key issues.
In conclusion, the whole layout and neatness of the paper do not leave too much objections, as it is prepared very carefully, but the quality of the discussion requires several amendments. Please answer all my questions and comments and attach the manuscript with marked changes. The objections presented by me do not undermine the quality of the paper, which will support in the further publishing process, certainly after careful consideration of my comments.
Author Response
Response to reviewers' comments:
“Line 51-56: It's good to characterize Monascus purpureus! “
The authors agree with the reviewer and appreciate the suggestion. Some characteristics of Monascus purpureus were added to the manuscript according to reviewer comments (Lines 38-47):
“Monascus is a xerophilic fungus which grows in a wide variety of natural substrates including rice and other cereals (Babitha et al 2006). The genus Monascus is divided into three species: purpureus, ruber and pilosus which are mainly isolated from oriental foods. The most important characteristic of this genus is their ability to synthesize pigments from polyketide chromophores and β-keto acids by esterification. The pigments produced by Monascus purpureus are classified into at least six types of pigments based on color; (1) red pigment (rubropunctamin, C21H26NO4, and monascurubramin, C23H27NO4); (2) orange pigment (rubropunctatin, C21H22O5 and monascorubrin, C23H26O5) and (3) yellow pigment (monascin, C21H26O5 and ankaflavin, C23H30O5). The structure of pigments produced by Monascus species depends on factors such as the type of substrate and nitrogen source, pH, temperature and agitation) (Dufosse, 2005, Haque et al., 2016, Kim et al., 2002).”
“Line 222-224: A very interesting observation, I think it needs a better discussion.”
The authors agree with the reviewer and following discussion was added to the manuscript (Line 231):
“This showed that Monascus purpureus did not synthesize cellulase to utilize the cellulose moiety of BSG. The rigid lignocellulosic structure of BSG also prevented the utilization of cellulose and therefore pigment production by Monascus purpureus.”
“Line 233-235: I understand, therefore, that the best growing medium is a liquid substrate.”
The reviewer is totally right since good fermentation characteristics were observed with dilute acid hydrolyzed medium, hence the best fermentation medium is a liquid substrate (Line 242-244)
“Line 251-255: Pretty confusing text regarding the speed of shaking.”
Lines 251-255 describes the effect of shaking speed of 250-400 rpm on pigment production. It was stated in the manuscript that at low shaking speed (250 rpm), large pellets were formed leading to mass transfer problems. High shaking speeds (400 rpm) caused high shear stress on the Monascus mycelium leading to low pigment production values. Hence, the highest pigment production was observed at moderate shaking speed of 350 rpm.
“Line 260-267: It is clear from the text that, in general, the method is quite unpredictable regarding the efficiency of pigment production.”
The effect of medium volume on pigment production was given in Lines 277-284:
25, 50 and 75 ml of fermentation media were used and the highest pigment production was observed with 50 ml of fermentation medium. Insufficient mixing was observed when pigment production was done in 25 ml of fermentation medium and this resulted in low pigment production. Vortex formation leading to poor oxygen transfer in 75 ml of fermentation medium was also observed which resulted in low pigment production values.
“Line 286-302: The text needs to be clarified, now it is quite unclear.”
The authors agree with the reviewer. The expression was corrected as suggested by the reviewer. (Line 305-314):
“The biomass and pigment concentration of Monascus mycelia is affected by the initial inoculum concentration. To observe the effect of inoculum concentration on pigment production, 50 ml of BSG fermentation medium was inoculated with 1, 2, 3 and 4 % (v/v) spore suspension solution, corresponding to 0.5×106, 1.0×106, 1.5×106 and 2.0×106 spores/50 ml of fermentation medium. The highest pigment production (22.25 UA500) was observed in the pretreated BSG medium inoculated with 2% (v/v) spore suspension. The pigments produced in the fermentation media inoculated with 1, 3 and 4% spore suspensions were 15.87, 18.15 and 11.36 UA500, respectively (Figure 3F). Our results showed that low inoculum ratio reduced the amount of biomass leading to lower concentration of pigment. However, high inoculum ratio yielded high biomass concentration that resulted in rapid consumption of nutrients in the fermentation medium required for pigment synthesis. “
“Line 362-365: These differences, however, are very large.”
The reviewer stated an important point that the difference in pigment production data in the literature were large. But the values given in the manuscript are obtained from the literature survey, from the published articles and the probable reasons of this variation were given in the manuscript as(Line 381-384):
“The variation in the literature data for Monascus pigment production may be related to several factors such as the strain of microorganism, the type of substrate and nitrogen source, the fermentation system, the method of pigment estimation, conditions used during fermentation.”
“Line 381-384: Probably? Certainly, these are the key issues.”
The authors agree with the reviewer. The word “probably” was removed from the manuscript according to reviewer comments (Line 400-402).
“This was due to the deamination of amino acids present in the medium by M. purpureus and the production of ammonia, which increased the pH of the fermentation broth. “

Reviewer 2 Report
The article is important for the production of natural pigments from plants, as the use of synthetic pigments may cause helath problems. I have some suggestions for authors to imrove their work. These follow the text sequence:
-Title
Monascus purpureus needs italics
-Abstract
Line 17.''..was achieved...''.
-Introduction
The Introduction section could be flourished by the incorporation of the health benefits of the Monascus pigments, claimed by the authors at conclusion.
Line 42.''...in the literature''.
-Materials and Methods
Line 63.''...in different solid medium:...''.
Line 84.''..the standard method 2540''.
Line 88.''The powder was directly placed...''.
Line 89.''..having a good...''.
Line 90. A dot is needed after the word ''sample''. Then the text could be read as: Four spectra of samples....''.
Line 109. ''The dinitrosalicylic method....''.
Line 118. Change ''are'' to ''were'', ''..were considered...''.
-Results and Discussion
Line 154. The comma is not needed: ''Carvalheiro et al.[20]....''.
Line 166. Change''can also be observed'' to ''were observed''.
Line 209.''In addition, the low pigment production....showed low...''.
Line 244.''In the present study,...''.
Lines 306-307. Change ''which was also stated'' to '' and has been cocumented by many authors''.
Line 344.''..which affect the pigmant formation...''.
Line 354.'', while afterwards was decreased''.
Line 365.''..and conditions used during fermentation''.
-Figure 4
The authors could provide Figure 4 with error bars.
Line 375.''..and then it was declined.''.
Line 381.''..was slightly decreased''.
Based on the aforementioned, I suggest a minor revision of the present work.
Author Response
Response to reviewers' comments:
-Title
“Monascus purpureus needs italics”
The authors agree with the reviewer. Title is corrected as demanded.
-Abstract
“Line 17.''..was achieved...''.”
The authors agree with the reviewer. The correction made.
-Introduction
“The Introduction section could be flourished by the incorporation of the health benefits of the Monascus pigments, claimed by the authors at conclusion.”
The authors agree with the reviewer. The health benefits of the pigments was added to the Introduction section as requested (Line 34)
“Line 42.''...in the literature''.”
The authors agree with the reviewer. The correction was made.
“Line 63.''...in different solid medium:...''.”
The authors agree with the reviewer. The sentence was changed to “in different solid(BSG): liquid(dilute sulfuric acid) ratios” Solid: BSG, liquid: dilute sulfuric acid
“Line 84.''..the standard method 2540''.”
The authors appreciate the correction. Missing article “the” was added to the sentence. The correction was made.
“Line 88.''The powder was directly placed...''.”
The authors agree with the reviewer. Missing article “the” was added to the sentence. The correction was made.
“Line 89.''..having a good...''.”
The authors agree with the reviewer. Missing article “a” was added to the sentence.
“Line 90. A dot is needed after the word ''sample''. Then the text could be read as: Four spectra of samples....''.”
The authors appreciate the correction. The sentence was corrected as requested.
“Line 118. Change ''are'' to ''were'', ''..were considered...''.”
The authors agree with the reviewer. The correction was made as claimed.
-Results and Discussion
“Line 154. The comma is not needed: ''Carvalheiro et al.[20]....''.”
The authors agree with the reviewer. The correction was made as requested.
“Line 166. Change''can also be observed'' to ''were observed''.”
The authors appreciate the correction. The sentence was corrected as requested.
“Line 209.''In addition, the low pigment production....showed low...''.”
The authors agree with the reviewer. The sentence was corrected as(Line 228):” In addition, low pigment production (3.07 UA500) was observed in BSG medium hydrolyzed with 1% (w/v) sulfuric acid.”
“Line 244.''In the present study,...''.”
The authors agree with the reviewer. Missing article “the” was added to the sentence. The correction was made.
“Lines 306-307. Change ''which was also stated'' to '' and has been cocumented by many authors''.”
The authors appreciate the correction. The sentence was corrected as claimed.
“Line 344.''..which affect the pigmant formation...''.”
The authors agree with the reviewer. The sentence was corrected as requested.
“Line 354.'', while afterwards was decreased''.”
The authors appreciate the correction. The sentence was corrected as requested.
“Line 365.''..and conditions used during fermentation''.”
The authors appreciate the correction. The sentence was corrected as requested.
-Figure 4
“The authors could provide Figure 4 with error bars.”
The authors agree with the reviewer. Error bars were added to Figure 4 as requested.
“Line 375.''..and then it was declined.''.”
The authors appreciate the correction. The sentence was corrected as requested.
“Line 381.''..was slightly decreased''.”
The authors agree with the reviewer. The sentence was corrected as requested.
